# Interpretable Clustering on Dynamic Graphs with Recurrent Graph Neural Networks

## Reproducibility Summary

**Scope of Reproducibility**

The main goal of the original paper is to perform dynamic node clustering in temporal graphs. The primary objective of this reproducibility study is to verify the major claim of the original paper stating that their proposed hybrid RNN (recurrent neural network) structures with graph convolutional networks (GCNs) outperform state-of-the-art graph clustering approaches. Another major claim of the paper is that under certain assumptions almost exact recovery of node-cluster membership estimations are achievable.

**Methodology**

The proposed models used in the original study utilizes a hybrid RNN-GCN model. RNN learns the approximation of decay rate using temporal graph structure information and GCN predicts the estimations of a node belonging to a certain cluster. In order to validate the claims, we have implemented the code using tensorflow deep learning framework and the code of the original paper using pytorch is available online. The simulation studies for the reproducibility study have been carried out on DELL ALIENWARE m15 R3 machine of an Intel core i7-10750H CPU @2.6 GHz equipped with 16 GB RAM and Windows 10 Home. This machine also has an NVIDIA GeoForce GTX 1660 Ti GPU with 6GB memory.

**Results**

The simulation results are inconclusive. Since the exact training and test data used by the authors of the original paper are not retrievable, the simulation results of the reproducibility do not always hold the claims of the original paper. As per the reported results from our reproducibility study, the baseline methods sometimes outperform the proposed models, even though the performance gap ($\leq 1\%$) is very low in a majority of the cases.

**What was easy**

The paper is very well written. The proposed models include sufficient algorithmic explanations to implement the code effortlessly.

**What was difficult**

The original paper does not include any explanation regarding the choice of hyperparameters. The number of simulation runs and any confidence interval on the performance metrics have not been explicitly specified in the paper either. The actual training and test data points used to report the results of the original paper are not retrievable. For these reasons, it becomes difficult to validate the claims and comprehend the overall semantics of the simulation results.

**Communication with original authors**

We had a suspicion that the original paper mistakenly reported the area under the curve (AUC) metric in place of F1-score and vice-versa. Hence, we had reached out to the authors of the original paper regarding some of our queries. The authors promptly responded and admitted that we were right. The authors also replied about the number of simulation runs used to report the results that was not earlier mentioned in the original paper.

# 1 Introduction

A common approach of interpreting graphs is to cluster nodes based on the edge connections. The wide variety of applications associated with graph clustering include: multi-scale community detection, information retrieval, data compression, social or biological network analysis, and so forth. Most of the existing literature studies focus on static graph clustering analysis, where the edge connections among the nodes do not change over time in a graph. However, the graph structures (e.g., edge association among nodes) evolve in practical settings. For example, the community association of users from social networks depends on varying factors, such as interests, occupation, and current location. Another example is the publication fields of researchers changing progressively in a citation network. Thus, the optimal clusters do not remain the same over time. Dynamic clustering analysis aims to track the evolving cluster memberships in graphs with the passage of time.

One of the critical challenges in a dynamic clustering problem is to determine the relative importance between historical graph formations and recently formed edges. Most often, graphs are subject to relatively slower changes. Hence, an approach completely disregarding historically formed edges or previous graph structures may overlook valuable information regarding majority unchanged cluster memberships. On the contrary, estimating equal relative importance for historical versus most recently formed edges can lead to slower convergence for classifying cluster memberships. Therefore, a proper balance of historical information has to be maintained to employ its value in predicting current cluster membership.

Previous studies ensure the balance of the historical effect by using a decay factor. Spectral clustering is a technique of decaying the factor by a constant amount over time steps to specify the weight of edges for estimating cluster memberships [1]. Recent efforts have been made to design various network models for learning the optimal decay rate at arbitrary time steps [5]. Moreover, LSTM (long-short-term memory) or RNN (recurrent neural network) structures in combination with graph convolutional networks (GCNs) have been applied to assess historical information without explicitly defining a decay rate [6]. However, such studies lack interpretability. The original paper [10] chosen for this reproducibility study aims to connect the previous works by proposing interpretable hybrid neural networks to approximately learn decay rates. The proposed semi-supervised clustering approaches are addressed as RNNGCN and TRNNGCN that combine the power of both RNN and GCN neural network architectures. The main purpose of the interpretable RNN layer is to capture the temporal dynamics of the graphs. Eventually, GCN layers are used to perform spectral clustering by utilizing the dynamics learned through the previous RNN architecture.

# 2 Scope of reproducibility

The primary goal of the paper is to build models for predicting the association of a node to a cluster in dynamic graph settings. Unlike most of the existing literature, this paper considers graph structures that evolve over time. The cluster membership can be modified over the time as well. The main difficulty emerges as deciding on the importance/weight factor of historical graph information and utilizing that information to calculate present node-cluster membership. Hence, the authors propose two hybrid RNN-GCN and transitional RNN-GCN architectures that can learn the importance of historical information of a graph through decay rate and apply the approximated decay rate to perform clustering. Following are the major claims of the paper that we will try to validate through the simulation studies conducted in this reproducibility study:

- Claim 1: The original paper claims that the proposed models RNN-GCN and TRNN-GCN outperform the state-of-the art graph clustering algorithms on majority of the datasets.

- Claim 2: The combination of RNN with GCN architecture can retrieve almost exact recovery of graph clusters considering the relative error to be $O(\frac{1}{n^{1/4}logn})$; where $n$ is the number of nodes in a graph.

- Claim 3: The performance of RNN-GCN continues to become worse as the number of classes/clusters increase in a dataset. However, TRNNGCN shows superior performance over RNN-GCN in such cases.

In order to test the first claim, we will apply the proposed models (RNN-GCN and TRNN-GCN) along with baseline methods (GAT, GCN, GraphSage, Spectral clustering, DynAERNN, and GCNLSTM) on both real datasets. Then, we will calculate the bounding error as per the input graphs using numerical analysis. Eventually, we will verify if the error of both the proposed classifiers remain under the theoretically defined bounding error to validate the second claim. Finally, we will observe the results along with progressively increasing the number of classes in the dataset and record the prediction ability of RNN-GCN and TRNN-GCN. As per the third claim, the performance of TRNN-GCN should be considerably better compared to RNN-GCN. Moreover, we will attempt to justify the same trend in case of real datasets.

# 3  Methodology

## 3.1  Model descriptions

The authors of the original paper propose two hybrid neural network architectures, RNNGCN and TRNNGCN. While RNNGCN considers a single decay rate $\lambda$, TRNNGCN focuses on utilizing a decay matrix $\Lambda$.

**RNNGCN** first attempts to learn the decay rate $\lambda$ using a RNN layer. Then, this architecture is followed by two layers of GCN to perform the actual clustering on the nodes of the graphs. The formal steps for this model have been recorded in Algorithm 1. At first, the algorithms takes the adjacency matrices $A_t \in \{0,1\}^{n \times n}$ over different time steps $t \in \{2, 3, ..., T\}$ as input, where $n$ is the number of nodes in a graph. Moreover, the algorithm requires $\Theta_T^{train} \in \{0,1\}^{n \times K}$ as input for training data. $\Theta_T^{train} \in \{0,1\}^{n \times K}$ refers to the cluster membership of node $n$ to a specific cluster $K$. In this case, $\Theta_{nk}$ becomes 1 if node $n$ belongs to cluster $k$, otherwise 0. The final outcome of the algorithm is to define the cluster membership estimates $\hat{\Theta}_T$ over the test data. The initial features are set as the identity matrix $I_N$ in line number 1 of the algorithm. In the same step, the algorithm starts by approximating the adjacency matrix as the adjacency matrix of the very first time step. For every other time step, the approximate adjacency matrix is updated by combining historical information with the present graph structure. The weight trade-off between historical and present graph information is determined by the $\lambda$ co-efficient, known as decay factor. This process has been mentioned in line numbers 3-4 of the algorithm. The next steps of the algorithm utilize two hidden GCN layers by applying $\sigma_1$ =ReLU and $\sigma_2$ =softmax activation functions. Next, the loss value is calculated on $(H^{train}, \Theta_T^{train})$ pairs using cross entropy function. Eventually, the trainable weights $W^1$ and $W^2$ are updated through the backpropagation process. The aforementioned steps are repeated for a discrete number of iterations. Finally, the nodes-cluster membership estimates $\hat{\Theta}_T$ are obtained by retrieving the highest confidence scores for every node against various clusters. It is noteworthy that the cluster labels are transformed into one-hot-encoded format for the prediction easement of the model.

---

**Algorithm 1:** RNNGCN

**Input:** $(A_1, A_2, ..., A_T)$: Temporal Graph Adjacent Matrices
$\Theta_T^{train}$: Training membership matrix
**Output:** $\hat{\Theta}_T$: Membership matrix estimates

1   $\hat{A} \leftarrow A_0, H_0 \leftarrow I_N$
2   **foreach** *iteration* $i \in \{1, 2, ..., I\}$ **do**
3      **foreach** *time step* $t \in \{2, 3, ..., T\}$ **do**
4         $\hat{A}_t \leftarrow (1 - \lambda)\hat{A}_{t-1} + \lambda A_t$
5      $H^{(1)} \leftarrow \sigma_1(\hat{A}_T H^{(0)} W^1)$
6      $H^{(2)} \leftarrow \sigma_2(\hat{A}_T H^{(1)} W^2)$
7      CrossEntropyLoss($H^{train}, \Theta_T^{train}$)
8      Backward()
9   $\hat{\Theta}_T \leftarrow \text{Onehot}(\text{argmax}_{(1 \leq j \leq n)} H_{jk}^{(2)})$

---

**TRNNGCN** is different from RNNGCN by considering a decay matrix $\Lambda$ instead of single co-efficient $\lambda$, as outlined in Algorithm 2. The notation $\Lambda \in [0,1]^{K \times K}$ defines the decay rate for a different pair of clusters/class labels. Another major difference of this model has been been specified in line number 4 of Algorithm 2. Here, the cluster membership estimates $\hat{\Theta}_{i-1}$ from previous iteration $i-1$ are utilized to learn the decay rates for current iteration $i$. Thus, a calculated cluster membership approximation $\hat{\Theta}_i$ from an iteration $i$ acts as an input for the next iteration $i+1$. The notation $\circ$ in line number 4 has been used to express element-wise multiplication. The rest of the steps in this algorithm are similar to the explanation provided for Algorithm 1. The primary intuition behind proposing the transitional version of RNNGCN (TRNNGCN) is the fact that various class labels may encounter heterogeneous optimal decay rates.

## 3.2  Datasets

The authors of the original paper considered five real datasets for conducting their experiments. Table 1 records various properties of the dataset. The four datasets (Reddit, Brain, DBLP-5, and DBLP-3) include node features for predicting the cluster membership. These datasets are used to evaluate the generalization capabilities of the proposed methodologies over the features of the nodes.

**Algorithm 2:** TRNNGCN

---

**Input:** $(A_1, A_2, ..., A_T)$: Temporal Graph Adjacent Matrices
$\Theta_T^{train}$: Training membership matrix
**Output:** $\hat{\Theta}_T$: Membership matrix estimates

1   $\hat{A} \leftarrow A_0$, $H_0 \leftarrow I_N$
2   **foreach** *iteration* $i \in \{1, 2, ..., I\}$ **do**
3      **foreach** *time step* $t \in \{2, 3, ..., T\}$ **do**
4         $\hat{A}_t \leftarrow (1 - \hat{\Theta}_{i-1}\Lambda(\hat{\Theta}_{i-1})^T) \circ \hat{A}_{t-1} + \hat{\Theta}_{i-1}\Lambda(\hat{\Theta}_{i-1})^T \circ A_t$
5      $H^{(1)} \leftarrow \sigma_1(\hat{A}_T H^{(0)} W^1)$
6      $H^{(2)} \leftarrow \sigma_2(\hat{A}_T H^{(1)} W^2)$
7      CrossEntropyLoss$(H_i^{train}, \Theta_i^{train})$
8      Backward()
9      $\hat{\Theta}_i \leftarrow \text{Onehot}(\text{argmax}_{(1 \leq j \leq n)} H_{jk}^{(2)})$
10   $\hat{\Theta}_T \leftarrow \text{Onehot}(\text{argmax}_{(1 \leq j \leq n)} H_{jk}^{(2)})$

---

| Dataset | Number of Classes | Time Steps | Nodes | Edges | Features | Dynamic Edges | Dynamic Class |
|---------|-------------------|------------|-------|-------|----------|---------------|---------------|
| Reddit  | 4  | 10 | 8921 | 264050 | 20 | Yes | No |
| Brain   | 10 | 12 | 5000 | 1955488 | 20 | Yes | No |
| DBLP-5  | 5  | 10 | 6606 | 42815 | 100 | Yes | No |
| DBLP-3  | 3  | 10 | 4257 | 23540 | 100 | Yes | No |

Table 1: Properties of the real datasets used for the performance evaluation

**Reddit** dataset is extracted from an American social news aggregation, discussion forum, and web content rating website. The nodes in this dataset correspond to different posts on the Reddit website [1]. An edge represents keywords connecting various posts. Hence, two nodes are connected via edges if they share the same keyword. The features relevant to nodes are derived by applying the word2vec mechanism on the comments of posts.

**Brain** dataset contains data regarding functional magnetic resonance imaging (fMRI) [2]. The nodes, in this case, are representatives of cubic brain tissues. Nodes are interconnected by edges in the case they share similar activation ratios in a certain time period. To achieve the node features, principal component analysis (PCA) has been applied on fMR scans.

**DBLP-3 and DBLP-5** contain data regarding bibliographic information for major computer science journals and conferences. All the data are collected from the DBLP website [3] starting from 2004 to 2018. Nodes refer to various authors. Edges form among nodes when the respective authors have a co-authored published manuscripts. Again, node features are generated by applying word2vec on the abstracts and titles of the papers. The class labels indicate the research fields of authors, such as, machine learning or networking, etc. The research fields/class labels are divided into three and five categories in case of DBLP-3 and DBLP-5, respectively. It is noteworthy that the research fields of the authors in these datasets remain static over the considered time duration. Finally, each time step refers to every year of publication.

All the datasets are publicly available for further use released by the authors of the original paper [4]. For experimental purpose, these datasets are randomly divided into 70% training/ 20% validation/ 10% testing.

### 3.3 Hyperparameters

All the methods used for the experiment purpose utilize two graph neural network layers as hidden layers. The size of the hidden layers are directly set as the number of classes in the respective dataset. For regularization, a dropout layer is used in between both of the hidden layers. The dropout rate has been selected as 0.5. Furthermore, an Adam optimizer has been chosen to optimize the loss function. The learning rate for training purposes has been set as 0.0025. Finally,

---

[1] https://www.reddit.com/

[2] https://tinyurl.com/y4hhw8ro

[3] https://dblp.org

[4] https://github.com/InterpretableClustering/InterpretableClustering

each neural network has been trained for 500 iterations in total to reach convergence. Unfortunately, the authors of the original paper do not explicitly state any reason behind choosing the aforementioned set of hyperparameters.

## 3.4 Experimental setup and code

Multiple baselines are considered to compare the performances of RNNGCN and TRNNGCN against the competent ones existing in the literature. For example, GAT [9], GCN, and GraphSage [4] baseline methods are supervised inhenrently that take node features into account for training. On the contrary, Spectral Clustering is an unsupervised method that disregards node features. Yet, all of the methods as mentioned above completely ignore historical information. DynAERNN [3] and GCNLSTM [2] factor in temporal information regarding both graphs and features throughout experiments.

The static methods (GAT, GCN, GraphSage, and Spectral Clustering) require a pre-processing step of normalizing the adjacency matrix of the graphs. Thus, the adjacency matrices of the input graph are accumulated and normalized at each time step. Then, clustering is performed on the resultant accumulated adjacency matrix. The other baseline models (DynAERNN, GCNLSTM, and EGCN) and proposed methodologies (RNNGCN and TRNNGCN) consider information regarding both temporal graphs and node features as input. We have implemented all of these baselines to verify their integrity.

For performance evaluation of each model, classical accuracy (ACC), F1-score (F1), and area under the ROC curve (AUC) classification metrics are used. Accuracy simply represents the state of correctness. F1-score is calculated from the harmonic mean of precision and recall. AUC refers to the ability of the classifier to distinguish class labels. The higher the performance metrics, the more reliable the performance of the models emerge. The ultimate goal of the experiment is performing node classification with temporal features. Even though, RNNGCN and TRNNGCN can not utilize node features, these take into account all the temporal historical information to approximate decay rate. This simulation study attempts to prove the applicability of proposed methodologies over real datasets DBLP-3, DBLP-5, Brain, and Reddit with temporal features. The code of the original paper is available online [5].

## 3.5 Computational requirements

The specific hardware infrastructure used by the authors of the original paper has not been explicitly mentioned in the paper. However, for the reproducibility study, the experiments have been carried out on DELL ALIENWARE m15 R3 machine of Intel core i7-10750H CPU @2.6 GHz equipped with 16 GB RAM and Windows 10 Home. This machine has a NVIDIA GeoForce GTX 1660 Ti GPU with 6GB memory. The original paper used pytorch for training the proposed models. This reproducibility study implements the paper using tensorflow to validate the variance sensibility of the models across different deep learning frameworks. The authors of the original paper have not included any report concerning required training time. As per our machine used for this reproducibility study, RNNGCN and TRNNGCN requires at most 1365 and 2074 minutes worth of training time for each dataset.

# 4 Results

We have implemented both of the proposed methodologies (e.g., RNNGCN and TRNNGCN) to verify the claims of the authors from original paper. Moreover, we have also implemented six state-of-the art algorithms (e.g., GCN, GAT, GraphSage, Spectral, DynAERNN, and GCNLSTM) for performance comparison. Then, the performance metrics have been recorded for four real datasets (DBLP-5, DBLP-3, and Reddit).

## 4.1 Verification of Claim 1

Table 2 indicates the averaged performance metrics obtained by the models over total time steps. This table records the comparison of the overall performance between the results stated in the original paper and the implementation done for this reproducibility study. From the results reported in the original study it is visible that their proposed TRNNGCN model outperforms other baseline models in most of the cases. However, according to the implementation of the reproducibility study, some of the baseline models outperform both RNNGCN and TRNGCNN. Nonetheless, it is noteworthy that performance gap between the proposed models and other best performing model according to our implementation is quite marginal ($\leq 2\%$) in majority of the cases. The exception in this case can be noticed in case of Brain dataset. In this case, their proposed methodology is outperformed by baseline model GraphSage significantly ($\geq 40\%$ performance gap). Thus, it can be concluded that the claim 1 (e.g., outperforming baselines) made by the authors of the original paper does not hold entirely.

---

[5]https://github.com/InterpretableClustering/InterpretableClustering

| Dataset | DBLP-3 | | | DBLP-5 | | | Reddit | | | Brain | | |
|---|---|---|---|---|---|---|---|---|---|---|---|---|
| Model | ACC | AUC | F1 | ACC | AUC | F1 | ACC | AUC | F1 | ACC | AUC | F1 |
| **Original Paper [10]** | | | | | | | | | | | | |
| GCN | 71.6 | 35.8 | 62.2 | 64.9 | 58.7 | 51 | 31 | 47.4 | 24.5 | 35.2 | 80.3 | 25 |
| GAT | 70.9 | 57.8 | 59.4 | 62.3 | 51.4 | 48.2 | 16.8 | 50 | 4.8 | 34.6 | 81.6 | 26.4 |
| GraphSage | 74.5 | 55 | 63.6 | 66.5 | 55.1 | 53.9 | 29.2 | 42.5 | 20.7 | **44.2** | **86.7** | 41.9 |
| Spectral | 45.7 | 51.2 | 51.6 | 43.8 | 51.3 | 45.6 | 30.1 | 51.7 | 24.1 | 42.7 | 68.1 | 41.7 |
| DynAERNN | 48.1 | 50.8 | 54.2 | 33.1 | 51.2 | 39.1 | 31.1 | **54.1** | **31.7** | 20.5 | 55.6 | 20.3 |
| GCNLSTM | 74.5 | 48.4 | 63.6 | 66.5 | 54.6 | 53.2 | 31.9 | 46.1 | 25.5 | 38.8 | 85.9 | 32.9 |
| RNNGCN | 75.9 | 66.7 | 68 | 65.7 | 58.6 | 55.4 | 33.6 | 49.7 | 20.5 | 41 | 84.7 | 38.6 |
| TRNNGCN | **78** | **73.8** | **72.1** | **67.4** | **63.5** | **57.9** | **33.6** | 53.2 | 25.6 | 43.8 | 85.7 | **42.4** |
| **Ours Implementation** | | | | | | | | | | | | |
| GCN | **78.34** | **89.12** | 69.45 | 68.5 | **87.67** | 56.31 | 29.24 | 55.93 | 18.75 | 21.12 | 67.62 | 12.56 |
| GAT | 78.17 | 87.76 | 68.81 | **68.74** | 86.97 | 56.67 | 31.85 | 55.92 | 15.54 | 39.81 | 82.6 | 33.18 |
| GraphSage | 77.56 | 86.46 | 69.77 | 66.5 | 80.49 | **58.9** | 28.8 | **56.37** | 16.38 | **64.93** | **91.29** | **91.29** |
| Spectral | 76.22 | 50.26 | 66.63 | 67.3 | 54.16 | 50 | **32.02** | 50.14 | 15.93 | 36.36 | 64.18 | 36.68 |
| DynAERNN | 45.69 | 51.57 | 52.34 | 37.36 | 51.06 | 41.63 | 29.16 | 52.44 | **28.98** | 26.28 | 58.61 | 26.01 |
| GCNLSTM | 77.48 | 86.5 | **70.56** | 67.68 | 84.57 | 57.66 | 31.23 | 56.7 | 20.93 | 41.52 | 85.1 | 40.1 |
| RNNGCN | 77.83 | 88.28 | 69.26 | 68.55 | 85.99 | 57.85 | 31.85 | 55.92 | 15.54 | 30.04 | 76 | 24.66 |
| TRNNGCN | 77.84 | 87.39 | 69.51 | 68.65 | 85.85 | 57.58 | 30.96 | 56.18 | 17.57 | 21.94 | 66.42 | 15.58 |

Table 2: Performance comparison of the proposed methodology (RNNGCN & TRNNGCN) against baseline methods averaged over timesteps

## 4.2 Verification of Claim 2

For the verification of the second claim, we performed some numerical simulations and matched the theoretical results with the experimental results stated in Table 2. Given the number of nodes in a graph being $n$, the relative error is stated to be $O(\frac{1}{n^{1/4}logn})$. However, we have been able to find a counter example, where the claim does not hold true. Considering that the number of nodes in Brain dataset is 5000, the theoretical relative error upper bound as per the claim of the authors should be around 3.21%. In contrast, the relative error based on simulation results from Table 2 for RNNGCN and TRNNGCN are 69.96% and 78.06%, respectively. Even, as per the results of the original paper, the hypothesis do not match with the empirical/simulation studies. We hypothesize this is due to the small size/nature of both the models and the datasets.

## 4.3 Verification of Claim 3

The claim 3 from the authors of the original paper states that with increasing number of node clusters in a graph, the performance of RNNGCN becomes worse but TRNNGCN can successfully maintain the performance. As per the results in Table 2, this claim is partially true. We can notice a significant deteriorating performance gap for Brain dataset, which has the high number of class labels (e.g., node clusters). However, TRNNGCN fails to maintain reasonable accuracy with the increasing number of node clusters as per the stated claim. As a matter of fact, RNNGCN performs even better than TRNNGCN according to the results of our implementation for this reproducibility study.

## 4.4 Results beyond original paper

Since the original paper did not report any confidence or error interval on the performance metrics, we have attempted to record the standard deviation of all the performance metrics over 5 simulation runs in Table 3. The main purpose of this experiment has been to test the variance sensibility of the proposed and baseline models across different training and test data points. It is noteworthy from Table 3 that often the proposed models show higher standard deviation averaged over different simulation runs. Moreover, the standard deviation of the proposed models increase significantly in case of complex datasets with higher number of class labels/node clusters, such as, Brain compared to baseline models. Hence, it can be said that the models can be highly sensitive towards various training/test splits.

# 5 Discussion

In this section, an overall justification of the easement and hurdles of the reproducibility study has been outlined.

| Datasets | DBLP-3 | | | DBLP-5 | | | Reddit | | | Brain | | |
|---|---|---|---|---|---|---|---|---|---|---|---|---|
| **Model** | **ACC** | **AUC** | **F1** | **ACC** | **AUC** | **F1** | **ACC** | **AUC** | **F1** | **ACC** | **AUC** | **F1** |
| **GCN** | 3.51 | 1.8 | 4.42 | 4.44 | **2.67** | **6** | 2.68 | 1.46 | 4.52 | 1.84 | 0.47 | 1.43 |
| **GAT** | 3.35 | 2.32 | 4.43 | 4.24 | 2.35 | 5.66 | 4.13 | 1.32 | 3.41 | 5.27 | 1.21 | 4.74 |
| **GraphSage** | 3.05 | 2.13 | 4.28 | 3.24 | 2.35 | 4.06 | 3.12 | **2.3** | 4.24 | 1.79 | 0.45 | 2.26 |
| **Spectral** | 0.34 | 0.12 | 0.1 | 0.03 | 0.11 | 0.01 | 0.03 | 0.02 | 0.07 | 2.27 | 1.23 | 1.84 |
| **DynAERNN** | 2.4 | 0.8 | 0.7 | **4.91** | 0.44 | 3.96 | 0.73 | 0.27 | 0.65 | 0.8 | 0.32 | 0.86 |
| **GCNLSTM** | **4.13** | 2.42 | **4.97** | 4.06 | 2.28 | 4.94 | 3.73 | 2.24 | **5.97** | 0.78 | 0.67 | 1.4 |
| **RNNGCN** | 3.24 | 2.35 | 4.06 | 3.76 | 2.03 | 4.44 | **4.15** | 1.31 | 3.45 | **8.15** | 9.12 | **9.97** |
| **TRNNGCN** | 3.3 | **2.43** | 4.59 | 3.82 | 2.35 | 4.89 | 3.52 | 1.12 | 3.01 | 7.6 | **10.42** | 9.07 |

Table 3: Standard deviation (in %) comparison of the proposed models against baselines

| Section | Information | Checked /Unchecked |
|---|---|---|
| Models & Algorithms | A clear description of the mathematical setting, algorithm, and/or model | ✓ |
| | A clear explanation of any assumptions | ✓ |
| | An analysis of the complexity (time, space, sample size) of any algorithm | ✗ |
| Theoretical claim(s) | A clear statement of the claim. | ✓ |
| | A complete proof of the claim. | ✓ |
| Datasets | The relevant statistics, such as number of examples. | ✓ |
| | The details of train / validation / test splits | ✗ |
| | An explanation of data that were excluded, and all pre-processing step. | ✗ |
| | A link to a downloadable version of the dataset or simulation environment. | ✓ |
| | For new data collected, a complete description of the data collection process, such as instructions to annotators and methods for quality control. | ✓ |
| Code | Specification of dependencies. | ✓ |
| | Training code | ✓ |
| | Evaluation code | ✓ |
| | (Pre-)trained model(s). | N/A |
| | README file includes table of results accompanied by precise command to run to produce those results. | ✓ |
| Experimental Results | The range of hyper-parameters considered, method to select the best hyper-parameter configuration, and specification of all hyper-parameters used to generate results. | ✓ |
| | The exact number of training and evaluation runs. | ✗ |
| | A clear definition of the specific measure or statistics used to report results. | ✗ |
| | A description of results with central tendency (e.g. mean) & variation (e.g. error bars). | ✗ |
| | The average runtime for each result, or estimated energy cost. | ✗ |
| | A description of the computing infrastructure used. | ✗ |

Table 4: The machine learning reproducibility checklist v2.0 [7] [8] by Dr. Joelle Pineau with respect to our reproducibility study

## 5.1 What was easy

The original paper is very well written. The motivation and model descriptions are well explained. Hence, we did not face any difficulty translating the algorithms into code implementation. Furthermore, the code of the original authors are available online that help us to verify all the hyperparameters whenever needed. The required libraries and their compatible versions have been mentioned in the documentation. The authors also made an effort to publish all the datasets utilized in their original paper. Thus, the datasets are easily accessible. It is also praiseworthy that the paper contains enough theoretical proofs for achieving a boundary on the optimal expected results beforehand.

## 5.2  What was difficult

First of all, the training, validation, and test data points originally used to report the results in the paper are not retrievable. In the code published by the original authors, the training-test splits are done randomly using the system clock as seed. Hence, the justification of the claims (e.g., the proposed methodologies outperform all the baselines) can be inconclusive, since the simulation results may vary depending on the contents of training and test dataset. For example, on same a dataset (e.g., DBLP-5) sometime GAT performs the best, while some other time GCN emerges as the best performing model due to having different training and test data points. We have found that the performance of the proposed models are highly sensitive towards the training and test data splits. Any confidence interval of the performance metrics have not been reported as well. The aforementioned reasons make it hard to test the major claim of the original paper (e.g., outperforming baselines) or verify the experimental results effortlessly. Furthermore, some pre-processing steps on the dataset (e.g., dropping disjoint nodes) have not been mentioned in the original paper. Essentially, this pre-processing step makes the training/test graph sizes considerably smaller than as reported in the original paper. In order to further facilitate the discussion section, we have outlined all the positive and limitation aspects of the reproducibility study in Table 4. Overall, we have inferred that availability to the access of exact training/test data, statistical significance testing, and considering adaptive overfitting over sufficient simulation runs are essential key points for any reproducibility study.

## 5.3  Communication with original authors

We initially detected that the authors of the original paper reported F1-scores in place of AUC metrics and vice-versa. Besides, the number of simulation runs have not been explicitly specified in the paper. Moreover, we wanted to have the exact training and test data points used by the original paper. Thus, we forwarded our queries to the authors of the original paper. They responded promptly to our email. They agreed that the AUC and F1-score misplacement identified by us was indeed right. Then, they replied to us about the number of simulation runs being 5 used to report the results in the original paper. However, they used random and non-retrievable seeds for training, validation, and test data splits for the original paper. Therefore, the exact training, validation, and test datasets are not possible to retrieve as per their code for any reproducibility study.

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
