# OpenReview forum: "Interpretable Clustering on Dynamic Graphs with Recurrent Graph Neural Networks"
_ML_Reproducibility_Challenge/2021/Fall — Reject_

### Official Review · Reviewer_1EPV · 2022-03-04
**Interpretable Clustering on Dynamic Graphs with Recurrent Graph Neural Networks**

**Rating:** 9
**Confidence:** 4

**Review:**

The paper is clear, with convincing details and provides a careful analysis and fair judgement of the reproduced results relative to the conclusions and claims made in the original paper.

---

### Official Review · Reviewer_xDqd · 2022-03-08
**Unsuccessful replication, but with limited exploration**

**Rating:** 6
**Confidence:** 4

**Review:**

This manuscript replicates results by Yao and Wong (2021), who introduce RNNGCN and TRNNGCN, two methods for clustering nodes in dynamic graphs. These methods attempt to estimate the rate of decay, either jointly or for each cluster, and are claimed in the original paper to outperform several baseline methods on real-world datasets.

In this replication study, the authors attempt to replicate the results on the same data sets. The authors discover that columns of AUC and F1 scores were swapped in the original paper. Even after correcting for this, the authors cannot reproduce the results of the original study: in some cases, baseline methods outperform the new methods. Moreover, the novel methods appear to be more variable (over 5 runs) than existing methods.  The authors also demonstrate that error bounds presented in the original paper do not always hold.  These data are potentially useful for potential users of the algorithms presented by Yao and Wang (2021).

However, the paper’s impact is blunted because the discrepancies between the original and replication are not investigated in detail. For example, several of the baseline methods require pre-processing to normalize the adjacency matrix, a potential source of variability between the studies. The results could be analyzed in more detail to determine if, as hypothesized, the *RNNGCN methods are more variable than existing approaches. In its present form, the paper contains a lot of tables with a lot of numbers; it would be helpful to reduce these down to summary statistics that can be more directly compared. The text could also use some light editing for style and grammar.

---

### Meta-Review · Area_Chair_eDQC · 2022-04-07

**Recommendation:** Reject
**Confidence:** 5

**Metareview:**

The paper is a strong reproducibility report, where they implemented additional baselines. However, reviewer xDqd note that the paper did not investigate the discrepancies between the original and replication in detail, especially on the issues of preprocessing. This makes the contribution borderline.

---

### Decision · Program_Chairs · 2022-04-09

Reject